# The Effects of Digitalization on the Sustainability of Small Farms

Kristina Šermukšnytė-Alešiūnienė [1,2] and Rasa Melnikienė [1,*]

1   Lithuanian Centre for Social Sciences, Institute of Economics and Rural Development,
    03220 Vilnius, Lithuania; kristina@agrifood.lt
2   AgriFood Lithuania DIH, 08412 Vilnius, Lithuania
*   Correspondence: rasa.melnikiene@ekvi.lt

**Abstract:** Digitalization of agriculture is one of the priorities of the EU's rural development strategy "From Field to Table", which promotes the creation of more added value and climate change mitigation in agriculture. A growing body of the literature argues that digitalization enables better information management, reduces production costs, and increases the potential for farm income growth, but only a few papers provide empirical studies on how digitalization improves the performance of small farms. To fill this gap in the literature, this paper presents a case study as empirical evidence of the impact of digital innovation on smallholder performance through a sustainable development lens. This paper reports research based on a pilot digitalization project implemented on a small organic farm. It examines the identification of logical links between the digitalization processes introduced and the impact of digitalization on the economic, social, and environmental performance of the small farm. The case study data were collected through semi-structured interviews and based on the results of a pilot project. The findings of this study provide evidence that the introduction of digital technologies has improved the economic performance of the farm, including a reduction in labor costs, improved customer relations, improvements in farmers' investment planning, and process redesign. Based on this study, recommendations are made to policymakers on how to promote the uptake of digital technologies in smallholder farming.

**Keywords:** digital agriculture; sustainability; small farm; digitalization effect; case study

## 1. Introduction

The digitalization of agriculture has received particular attention from researchers in recent years, with a growing number of studies summarizing successful digitalization experiences, looking for success factors in this process, and addressing future research priorities [1,2]. Research in this area is needed by policymakers implementing evidence-based policies. Findings from the evaluation of successful experiences and regularities have implications for further practical digitalization processes and generate significant societal impacts for agriculture. International organizations have also placed a strong emphasis on digitalization in their agricultural reviews [3].

Digitalization encompasses a wide range of phenomena and technologies such as apps on mobile phones, sensors, big data, the Internet of Things, robotics, ubiquitous connectivity, systems integration, artificial intelligence, machine learning, digital twins, and blockchain [4].

Digital transformation is defined using the following three distinct elements:

1.   "Technological (use of new digital technologies such as social media, mobile, analytics, or embedded devices)
2.   Organizational (a change in organizational process or the creation of a new business model)
3.   Social (a phenomenon that is influencing all aspects of human life, e.g., enhancing customer experience)" [5].

The interest of researchers in the digitalization of agriculture is driven by the complexity of the phenomenon and its multi-dimensional impact on the development of society. Digitalization implies that management tasks on-farm and off-farm (in the broader value chain and food system) focus on different sorts of data (on location, weather, behavior, phytosanitary status, consumption, energy use, prices, economic information, etc.). The data obtained are used to interpret the past and predict the future, to make more timely or accurate decisions, through constant monitoring or specific big data science enquiries. Digitalization is thus perceived as a transformative force in agricultural production systems, value chains, and food systems [6] and is linked to the creation of higher economic added value, social change in rural areas, and the achievement of climate change mitigation targets in agriculture. In this context, digital technologies lay the foundations for a sustainable agricultural future [7–9]. In addition to the positive aspects of this phenomenon, researchers warn of the threats posed by digitalization if this process is only available to large industrial farms. A study published in 2015 by Poppe, K. et al. noted that "information and communication technology (ICT), combined with higher food prices and demographic changes could fundamentally shift the competitive advantage from family farms to more industrial holdings, leading to radical structural change in agriculture" [10].

To achieve a breakthrough in digitalization and massive involvement of farms, one of the problematic issues is the digitalization of small farms. In many countries, small farms are still an important part of food systems [11–14]. The attitude toward small farms as inefficient, which dominated for a long time, has changed dramatically in recent years, and the attention of policymakers and society towards small-family farming has risen significantly [11]. Society's awareness of the importance of sustainable development has revealed the potential of small farms to address new challenges related to food security and diversity, maintaining the social viability of the countryside, and the environmental benefits of traditional farming [12,13,15,16]. The European Union is no exception, as it has often asserted that small family farms are the "backbone of European farming". At the same time, however, small farms are typically associated with low incomes. They lack the resources to introduce innovative technologies [11].

An analysis of scientific references shows that in many studies, when the digitalization of small farms is included in the field of research, farm size is analyzed as a factor for innovation [17–20]. Studies focusing on the evaluation of the digitalization of small farms conclude that small farms are left behind in the digitalization process [21–24].

Researchers have identified not only limitations to digitalization on small farms, such as a lack of skills or financial resources but also a lack of motivation to adopt digital technology, with farmers doubting its benefits [22,24]. As farmers become more active in digital innovation, they are looking for an answer to the question of what the benefits and impacts of farm digitalization are on farms, food supply chains, and food systems, as well as on rural communities. For example, some researchers argue that digital technologies themselves do not generate economic benefits, they only allow them to be strengthened [25]. Therefore, researchers studying digitalization processes are exploring how to improve access to innovative technologies for small farms [5,16,26] and to encourage them to adopt digital innovations [22,27].

A review of the scientific literature shows that there are not many studies that show the benefits of digitalization for small farms, and researchers have identified a need for such studies [22]. Our study seeks to answer this question based on a sustainability paradigm that emphasizes the economic, social, and environmental aspects of smallholder farming.

This study aims to investigate the links between digitalization and sustainable development in small farms and to identify which economic, social, and environmental criteria best reflect the changes driven by innovation. This study was carried out on a small farm where a pilot digitalization project was implemented, which allowed the establishment of evidence-based logical links between the technologies introduced and the impact of digitalization on the small farm's economic, social, and environmental performance.

This paper is organized in four sections as follows. The introduction of this paper justifies the relevance of the problem and provides an overview of research on the topic. The second section presents the theoretical framework and research methodology. It indicates the link between digitalization and sustainable development, explains the case study approach, examines sustainable development indicators, and provides a list of indicators for measuring changes in farm performance. The third section presents the results of the semi-structured interviews and interprets them as evidence in the context of the research problem. The last section of the paper focuses on the scientific discussion, the critical analysis, and comparison of the results of this study with other studies and the overall conclusions.

## 2. Materials and Methods

### 2.1. Theoretical Framework

The sustainable development paradigm was chosen to assess the effects of digitalization on farm activities, as this theoretical approach is still an important part of agricultural development research [28,29]. Sustainable development is defined as a multidimensional concept that includes three key aspects: economic, social, and environmental [30]. The relationships among these dimensions are generally assumed to be compatible and mutually supportive [29].

Since the definition of the concept of sustainable development, it has become an important theoretical approach for assessing various processes in agriculture [31–33]. This methodological approach is also used for studies that analyze the experience of agricultural digitalization. Academic and popular literature on digital agriculture identifies a link between precise information gleaned from big data and environmentally sound management. This link is often presented as so profound that it represents a paradigm shift from production-based agricultural goals to sustainability [4]. Digitalization is associated with increased competitiveness of companies in global markets, as well as sustainability and better management of the territories [34]. Digital technologies can help to raise on-farm productivity, improve resource use efficiency, and support climate resilience [35]. International organizations have set targets for digitalization to make the agri-food sector more efficient, inclusive, and environmentally sustainable, thereby increasing benefits for farmers, consumers, and society at large [36].

The recognition of sustainable development as a fundamental goal of society's evolution has led to the focus of research on indicators and indicator systems to measure, monitor, and manage changes in sustainable development at different levels, considering the needs of decision-makers and researchers [37]. Relevant to our study are findings that identify several problems in applying indicator systems at the enterprise level. Researchers argue that a sustainability indicator framework, when developed at the company level, needs to consider the company's strategy and that the indicators need to provide an integrated view of the company's performance [37].

In designing an indicator system to measure the effect of digitalization on the sustainable development of small farms, we considered the experience of constructing such systems in small businesses. The concept of sustainable development is often used as a synonym for environmental protection in the formulation of environmental challenges for countries, regions, and businesses [28,29,37]; in the case of small businesses, the definition of sustainable development emphasizes the economic and social dimensions. Experts combine economic performance with social and natural performance to form a balanced scorecard, stressing sales and profit [38].

Research on small business sustainability indicator frameworks shows that sustainability from a business perspective is not just about doing green business or just paying attention to the natural environment, but also about how a business strategy is implemented in the context of economic and social sustainability [29,38]. An analysis of how economic aspects influence a small company's sustainability shows that there is a correlation, which leads to the conclusion that financial performance promotes sustainability practices [39].

In 2023, Gumbi, N. et al. published an article "Systematic Literature Review of Sustainable Digital Agriculture for Smallholder Farmers Research", which showed that researchers also pay particular attention to economic performance when assessing changes in sustainability on small farms, describing them as "benefits of using digital solutions" [22]. The economic indicators mentioned in this study are productivity, better earning and higher yield, improving product quality, lowering costs, or reducing transactional costs, access to markets, resilient farm production, suppliers, and value chain, improving farming decisions and predictions, cash management, etc. The systematic review also revealed efficiency, sustainability, transparency, a positive impact on agricultural income, and real-time data.

The study was based on the Farming Systems Research (FSR) approach, which examines the effectiveness of modern technologies on farms by looking at farmers' practical experience. Researchers who have summarized the experience of using this research suggest the following:

- "Drawing on farmers' experiences is the best way to inform researchers if the aim is to produce applied research results.
- Farmers' experiences can be of significant assistance to the research being carried out in their farm environment.
- Researchers can consider the diversity of farming patterns.
- Researchers can provide farmers with new knowledge during the study" [40].

Farmers can analyze their farming practices as part of the research.

### 2.2. Research Methodology

The research methodology presented in this paper was chosen because of the research problem and the questions posed, which are still empirically new and theoretically poorly defined. An analysis of the academic literature shows that the digitalization of small farms is a slow process, and successful examples of digitalization of small farms are not often found in practice and are the subject of research.

Many experts in scientific methods emphasize the appropriateness of the case study approach when the research problem is unexplored and original, as case studies allow for an in-depth assessment of the phenomenon and the development of new theoretical assumptions [41–43]. The case study is a research strategy that focuses on understanding the dynamics present within single settings [41]. Case studies can involve either single or multiple cases and numerous levels of analysis. Case studies typically combine data collection methods such as archives, interviews, questionnaires, and observations. The evidence may be qualitative (e.g., words), quantitative (e.g., numbers), or both.

A qualitative case study methodology enables researchers to conduct an in-depth exploration of intricate phenomena within some specific context [44]. The case study approach requires that the characteristics of the farm be defined to determine the scope of the results [41].

Based on existing case selection strategies in scientific research, purposive sampling is normally used when the case is selected for a specific objective and not because it has some elements of the phenomenon under analysis [45,46]. In response to the research question, we based the farm's selection in this study on purposive sampling according to Patton [47], using the following case selection strategies:

(a) an intensity strategy, where the case that provides the most information is selected
(b) a typical case strategy, which determines the economic group of farms represented by the farm
(c) a theoretical strategy, which describes the features of the farm's business model that are relevant for the interpretation of the survey results.

The farm under study was in line with the intensity strategy because of participation in the EU-funded study "Business Digitalization toolbox. Smart Industrial Remoting: remote working in non-digitalized industries—Pilot Project" [48] and had implemented a digitalization pilot project. The pilot project aimed to evaluate the principles of digitalization's

good practices in small farms in real-world settings and to develop recommendations in the toolbox on how to successfully digitize small farms. A farm management system was implemented, consisting of the modules listed in Table 1. Digitalization includes the following processes:

1. Resource and product stock management;
2. Crop and harvest planning;
3. Agricultural production;
4. Recipe development;
5. Processing of agricultural products;
6. Sales;
7. Accounting and reporting to public authorities.

**Table 1.** Modules of the digital farm management system.

| Module | Function |
|---|---|
| Environmental observation | The system's integrated real-time weather forecasting allows farmers to make decisions based on current and predicted weather patterns, reducing risk. |
| Crop and pest management | Pest and disease management is based on image recognition technology, which helps to detect pests and diseases early and intervene to preserve crop health and reduce losses. |
| Technology integration and data analysis | Sensors are used to measure certain soil parameters, and for monitoring equipment and machinery integrated with IoT devices. The analysis and reporting function uses these data to generate performance indicators and support future agricultural decision-making. |
| Market dynamics and financial integration | It provides real-time market prices and ensures that farmers receive a fair price for their produce. |
| Integrated farm management | The farm was able to monitor and record crop cycles, costs, income, and labor inputs. Task scheduling ensured that activities such as planting, irrigation, and harvesting were carried out at the optimum time, thus streamlining operations, and reducing waste of resources. |

Through the project, the farmer has improved his technological and business management knowledge and, therefore, has a distinctive and unique experience that allows him to apply digitalization most effectively.

By implementing a typical case strategy, the farm under study met the quantitative and qualitative criteria of a small farm, both in theory and practice. Firstly, we followed the approach that the physical size of a holding is globally the most commonly used criterion to define small farms. Commonly, studies define small farms as "those with less than either 5 or 2 hectares of cropland" [11]. On the other hand, some academics argue that physical size is not enough to define a small farm but that the socio-economic context in which they operate, such as market and infrastructure conditions, is also important. We considered their proposed classification of farms into three categories, depending on their economic objectives and the level of market integration: small farmers, farmers in transition, and subsistence-oriented small farmers. We conducted the study on a farm with 0.5 ha of land, 2 employees, and only family members. Depending on the degree of market integration of the farm, the farm could be described as "Commercial small farmers run their farms as businesses", while commercial agriculture was an important source of income for them [16].

According to the theoretical strategy, the farm business model is oriented to environmental and social sustainability [49]. Farmers are developing a biodynamic farm. The farm has a high plant diversity with around 50 plant species. The farm not only produces agricultural products but implements the whole food supply chain from field to fork and

has a website for consumers. The farm interacts directly with customers, delivering baskets of vegetables, fruit, and other produce to the homes of customers who are ready to pay a higher price for an exclusive product. We were, therefore, able to assess how well the agricultural production process management system contributes to the sustainability of the farm's activities, not only in the primary sector but also in the other stages of the food chain: processing and marketing. Poor-looking production or production that cannot be sold fresh on the market is processed and sold to customers as jams, sauces, steamed cold vegetables salads, etc. The farm is implementing circular economy principles, using bio-waste to make compost, with the aim of not throwing away produce. This special methodology is being created on zero-waste small farm productivity and maintenance.

To summarize the case study strategies, the farm we studied can be characterized as a small commercial farm focused on environmental and social sustainability, with digital management system experience. The farm is managed by a woman, but the equal opportunities aspect was not included in the study.

We used semi-structured interviews as the main data collection tool for the case study. Interviewing has become one of the most commonly used methods of collecting data in qualitative research studies, with the 'one-to-one' interview arrangement predominantly used within a semi-structured format [50].

In the preparatory stage, we prepared interview guides to structure the answers according to sustainability areas, indicators, and processes. Based on the analysis of the scientific literature, a list of sustainable development indicators was drawn up to measure the effects of digitalization (Table 2). The indicators were chosen in such a way that farmers could provide answers based on their knowledge and information.

**Table 2.** Sustainable development indicators for small farms.

| Economical | Social | Environmental |
|---|---|---|
| E.1. Harvest | S.1. Farmer's labor costs | |
| E.2. Income | S.2. Opportunities to combine farming with other activities | A.1. Amount of food loss |
| E.3. Quality of production | | A.2. Fuel consumption |
| E.4. Productivity or efficiency | S.3. New skills transferable to other activities | A.3. Use of chemical fertilizers and pesticides |
| E.5. Cashflow | S.4. Job satisfaction | A.4. Water and soil pollution |
| E.6. Marketing costs | S.5 Balance between work and family interests | A.5. Biodiversity |
| E.7. Intermediate consumption | | |

Our study aimed to empirically assess changes in farm performance one year after the pilot project.

## 3. Research Results

The interview consisted of two steps. The first part aimed to identify the primary effect of digitalization, as the farmer expected to see the results of the introduction of new technologies soon after their introduction. In the second part of the interview, the farmer was asked to assess the new possibilities for sustainable development that the primary results would create for the farm and how these would be transformed into lasting changes in the farm's activities.

To identify the primary effects of digitalization, the farmer was asked to describe the changes that have taken place on the farm based on a list of indicators. She was asked to describe the economic, social, and environmental effects generated by each digitized process. A matrix table was filled in, linking the effects to the digitized processes. This way, information was gathered on which processes generated the highest number of effects and which effects were mentioned as the most repeated (Table 3).

**Table 3.** Primary digitalization effects by process.

| Process | Economical | | | | Social | Environmental |
|---|---|---|---|---|---|---|
| | E.1. Increase in Harvest | E.2. Increase in Income | E.3. Increase in Quality of Production | E.5. Increase in Farm Efficiency | S.1. Decrease in Farmer's Labor Costs | A.1. Decrease in Agricultural Production Losses |
| 1. Resource and product stock management | | X | | X | X | X |
| 2. Crop and harvest planning | X | X | | X | X | X |
| 3. Agricultural production | | X | X | X | X | X |
| 4. Recipe development | | X | | X | X | X |
| 5. Processing of agricultural products | | X | | X | | X |
| 6. Sales | | | | X | X | X |
| 7. Accounting and reporting to public authorities | | X | | X | X | |

The farmer prioritized economic indicators to describe the impact of digitalization. The economic effects of digitalization, according to the farmer's perception, were reflected in increased yields and incomes, improved quality of production, and increased efficiency from the list of indicators shown in Table 2.

This shows that for a farm whose business model is oriented towards environmental and social sustainability, it is important to generate enough income to continue the farm's activity and to guarantee the economic sustainability of the activity. The digitalization of agricultural production and crop and harvest planning had the highest number of effects, with five indicators chosen by the farmer to describe positive changes, digitalization of resource and product stock management, and digitalization of recipe development processes, with four indicators each, respectively. Changes in processing, marketing, and digitalization of accounting and reporting to public authorities were identified as the three indicators to describe effects.

Production efficiency increased as a result of the digitalization of all farm processes. Efficiency gains were made in terms of increased farm income due to better harvest and crop rotation planning, the availability of information on weather changes and disease prevention, and the ability to plan agricultural work accordingly. The farmer noted the importance of information on losses from natural disasters to avoid such losses. The ability to optimize the processing of products that farmers could not sell fresh to consumers and to develop recipes based on stocks of such products, also increased income.

The farmer mentioned that she does not aim to increase yields with an organic farm, but that the digital tool for crop and harvest planning helps them to increase harvest by providing information on the harvest of the plants on specific plots, considering the previous crops, and by better planning of the crop rotation and production technology. The digitalization of accounting made it possible to assess the profitability of production. Farmers can plan to review their product mix and eliminate unprofitable products.

In terms of social sustainability, the most important effect was the reduction in the family labor costs. It was present in all the digitized business processes, except processing (Table 3). The most important beneficial change in terms of labor costs was related to accounting and reporting to public authorities. Digitalization made it possible to automate the preparation of many reports to public authorities, reducing the time needed for this work by several times and reducing the risk of mistakes.

From an environmental point of view, the most important indicator for the farmer was the reduction in agricultural production losses, which is in line with the farm's ambition to implement a zero-waste strategy.

The farmer said that the introduction of digitalization had not changed many of the environmental indicators of their farms: fuel use, fertilizer use, soil, and water pollution. She stressed that environmental issues are irrelevant because the family farms organically. She also did not identify social indicators among the effects indicators that are important: opportunities to combine farming with other activities, new skills that could be applied in other activities, job satisfaction, and balance between work and family interests.

In the second part of the interviews, the farmer was asked to describe whether the changes taking place on the farm are long-term and create new opportunities for sustainable development. During the interviews, for each of the indicators, the farmer formulated new opportunities in economic, social, and environmental terms (Table 4).

**Table 4.** Potential for sustainable development of small farms through digital technologies.

| Indicator | Economical | Social | Environmental |
| --- | --- | --- | --- |
| E.1. Increase in harvest | Higher harvest leads to increased customer numbers and increased revenues. | Increasing the supply and variety of local food for consumers. | Sustainable technologies increase harvest without increasing chemical pollution. |
| E.2. Increase in income | Increasing income provides the opportunity to invest in equipment and inventory. | Farmers' quality of life improves. Better equipment improves working conditions. | Investments can be made in storage facilities to reduce agricultural product losses. |
| E.3. Increase in quality of production | Improving the quality of the produce allows it to be sold at a higher price. | Increasing the supply and variety of local food for consumers. | Reduces the amount of agricultural production that can be thrown away. |
| E.5. Increase in farm efficiency | The efficiency of production types is evaluated, and unprofitable production is eliminated. | The competitiveness image of small organic farms in society is strengthened. | The economic efficiency of environmental solutions can be calculated. |
| S.1. Decrease in farmer's labor costs | It provides additional time to develop the farm. | Better balance between professional and family interests. Opportunities to combine farming with other economic activities. | Saved time can be invested in processing produce and reducing agricultural production losses. |
| A.1. Decrease in agricultural product losses. | Farm income increases because of reduced production losses. | Positive image of the farm in the community and among consumers. | Reduces the amount of agricultural production that can be thrown away. |

The farmer reviewed the initial changes in the farm following digitalization as creating long-term opportunities to develop the farm, increase income, and invest in technology. At the same time, the economic changes created additional social opportunities for the farm, both within the farm and in the relationship with the consumer community. On the other hand, environmental effects, such as the reduction in agricultural production losses, create new economic and social opportunities. During the interview, the farmer

emphasized that digital technologies have linked all on-farm processes into a consolidated, more manageable, and more resource-efficient system.

## 4. Discussion and Conclusions

Successful digitalization of agriculture is not possible without the involvement of small farms, because small farms represent a significant share of producers in many countries around the world. The literature review has shown that the process of digitalization of small farms is slow and faces significant challenges, such as low digital skills of farmers, lack of funds for the acquisition of technology, and lack of motivation to adopt digital technologies [24,27].

In this study, we focused on the interests of farmers to empirically identify the benefits of digitalization for small farms to deal with the issue raised in the scientific literature that "digital technologies must be useful enough to provide a benefit to the farmers, either through an improvement, by doing something easier or cheaper than before, or an innovation, something that was not previously done because of financial constraints or an incongruence between the technology and farmer's skills" [17].

Our study seeks to develop the knowledge that explains the interest of small farms in participating in the digitalization process. Its results support the finding in the scientific literature that farmers need to know what benefits they can expect from the adoption of digital technologies. In parallel, our results confirm the hypothesis put forward in other studies that the factors limiting digitalization on small farms, which depend on farmers' decisions, are easily eliminated by the arguments of economic benefit. After learning about the possibilities and benefits of technology, smallholder owners seek to develop new skills and allocate resources to purchase equipment [22]. The method used for the research, case study, follows the trend of methods used in agricultural digitalization research. Given the lack of statistical data on digitalization, surveys and case studies are the most used methods to collect empirical data in farm digitalization research [18,21,22,50].

In our study, the use of a case study approach allowed us to thoroughly assess the phenomenon of smallholder digitalization and to generate new theoretical insights, as recommended by academics who have developed this approach [42–44].

As a case study, a small farm was chosen where a digitalization pilot project was implemented a year ago. The farm was able to provide us with a lot of knowledge about the impact of digitalization on the farm's operations, as we had information about the digital solutions implemented during this pilot project. During the study, we were able to compare the performance of the farm before and after digitalization and to identify differences based on information collected empirically. The short period between the introduction of digitalization technologies and the interviews eliminated the risk that the farm's performance would change due to market conditions. We also avoided the limitations inherent in many studies that have used the interview method [22,51,52], where respondents answer questions about the benefits of digitalization, but their answers are not validated in terms of farmers' experiences with specifically digital solutions. This, therefore, leaves unanswered the question of whether expected or real benefits were measured.

As our analysis of the scientific literature shows, researchers looking at the motivations for the digitalization of small farms rely mainly on economic indicators to define the benefits of digitalization [22]. Some studies propose to define the requirements of small farms for digital technologies, very specifically in terms of the cost of the equipment and the operational cost: lower cost, low power, and ruggedized theft-proofing [27]. In other words, many researchers propose that the benefits of digitalization of small farms should be seen only from an economic perspective.

The sustainable development paradigm is inherent in research on the digitalization of small farms [3,7,9,22]. Using the paradigm of sustainable development as the theoretical basis for this study, we carried out an integrated assessment of digitalization, identifying the economic, social, and environmental effects of digitalization on the farm. At the same time, we tested the previously mentioned hypothesis that farmers prioritize economic

indicators when assessing the benefits of digitalization. Our study shows that even when selecting a small farm with a socio-environmentally driven business model, the farmer prioritized economic indicators. We concluded that in small farms, the data collected and processed on an operational basis through digitalization provide farmers with important information for making business decisions, which allows them to improve the economic performance of the farm. Our study confirmed the findings of other (non-agricultural) small business sustainability studies that economic performance is the most important factor in small businesses and that successful economic performance improves social outcomes and creates opportunities for environmental solutions [39].

Based on the scientific literature analysis, a list of sustainability indicators was compiled, where each indicator is traditionally assigned to one of the sustainability domains. However, the interviews also examined the complexity of the indicators, as it is generally assumed that the links between these dimensions of sustainability are compatible and mutually supportive [29,31]. The results of this study showed that economic, social, and environmental effects are interconnected. For example, the food loss indicator, although classified as a group of environmental indicators in the scoring systems developed in the scientific literature, was also used as a basis for the farmer to define the economic and social effects of the farm. On a small farm with only owner-operators, the social effect of reduced labor costs was important for the farmer. However, the farmer also linked this indicator to better economic results: additional working time to run the business.

In our research, we had to deal with the limitations of the lack of economic data on farms that have adopted digital solutions and the limited capacity to use quantitative methods for evaluation. We used a case study for the research, and the farm under study was selected through purposive sampling using the case selection strategies. Based on these strategies, the findings of this study can be applied to small commercial farms. This study empirically confirms the existence of economic, social, and environmental effects in our case study, substantiates the link between digitalization and farm performance, and provides new arguments for further research on the digitalization of small farms. The results of this study contribute to social science research that identifies farm digitalization as a priority issue in the recent academic literature [2,3,22].

This study did not evaluate agricultural policy decisions to promote digitalization, but the results suggest two recommendations for policymakers. Firstly, digitalization pilot projects are an important instrument for presenting the successful digitalization experiences of small farms, and support for such projects could be an accelerator for small farm digitalization. Secondly, the experience of digitally skilled farmers should be used to advise and train other farmers. The results of this study can be presented for discussion with agricultural advisors and farmers' associations.

**Author Contributions:** Conceptualization, R.M.; Methodology, K.Š.-A. and R.M.; Validation, K.Š.-A.; Formal analysis, K.Š.-A.; Investigation, K.Š.-A.; Data curation, K.Š.-A.; Supervision, R.M.; Project administration, R.M. All authors have read and agreed to the published version of the manuscript.

**Funding:** This research received no external funding.

**Institutional Review Board Statement:** Not applicable.

**Informed Consent Statement:** Informed consent was obtained from the subject involved in the study.

**Data Availability Statement:** Data is contained within the article.

**Acknowledgments:** The research was carried out on a farm where a pilot project has been adopted in the framework of the EU-funded study "Smart Industrial Remoting: remote working in non-digitalized industries". The study is implemented by PPMI. Contract number: LC-01796763.

**Conflicts of Interest:** Author Kristina Šermukšnytė-Alešiūnienė was employed by the non-profit organization AgriFood Lithuania DIH. The remaining authors declare that the research was conducted in the absence of any commercial or financial relationships that could be construed as a potential conflict of interest.

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
