# Peer review of "The Effects of Digitalization on the Sustainability of Small Farms"

_sustainability, doi:10.3390/su16104076_

Round 1

Reviewer 1 Report

Comments and Suggestions for Authors

Dear Author(s),

Article is well structured and the topic is interesting. However, following comments should be addressed prior to further processing of the article.

1)      Refer to abstract: Abstract is poorly written. Needs to be rewritten.

2)      Refer to section 1: Last paragraph of section 1 should describe structure of the article i.e. missing in the article.

3)      Refer to line # 48: Poppe and co-authors or Poppe et. al.? Recheck the whole article for such typos.  

4)      Refer to line # 57: … attention of policymakers and society towards small-family farming has risen significantly… Include reference here.  

5)      Refer to line # 67: Recheck it [17,18,19;20] [41;42,43] for typos.

6)      Refer to section 1 and 2: There is insufficient discussion on digitization. Authors need to discuss it in more detail as it is the core topic of this study.

7)      Refer to whole article: How sustainable development is linked with digitization?

8)      Refer to section 1: What is the criteria to define big farm and small farm?

9)      Refer to line # 127: Authors need to explicitly mention “benefits of suing digital solutions”

10)  Refer to line # 144: A single sentence?

11)  Refer to line 3 161: Is Patton the name of author or the name of strategy?

12)  Refer to line # 168: Include reference for direct-to-consumer business model. Moreover, what are other business model and why this model is selected by the authors?

13)  Refer to line # 183: Authors need to confirm the title “Smart Industrial Remoting: remoting working in non-digitalised industries”.

14)  Refer to Table 1: Authors have mentioned about weather pattern prediction in environmental observations. Where are the further details?

Good luck.    

Comments on the Quality of English Language

Dear Author(s),

Article is well structured and the topic is interesting. However, following comments should be addressed prior to further processing of the article.

1)      Refer to abstract: Abstract is poorly written. Needs to be rewritten.

2)      Refer to section 1: Last paragraph of section 1 should describe structure of the article i.e. missing in the article.

3)      Refer to line # 48: Poppe and co-authors or Poppe et. al.? Recheck the whole article for such typos.  

4)      Refer to line # 57: … attention of policymakers and society towards small-family farming has risen significantly… Include reference here.  

5)      Refer to line # 67: Recheck it [17,18,19;20] [41;42,43] for typos.

6)      Refer to section 1 and 2: There is insufficient discussion on digitization. Authors need to discuss it in more detail as it is the core topic of this study.

7)      Refer to whole article: How sustainable development is linked with digitization?

8)      Refer to section 1: What is the criteria to define big farm and small farm?

9)      Refer to line # 127: Authors need to explicitly mention “benefits of suing digital solutions”

10)  Refer to line # 144: A single sentence?

11)  Refer to line 3 161: Is Patton the name of author or the name of strategy?

12)  Refer to line # 168: Include reference for direct-to-consumer business model. Moreover, what are other business model and why this model is selected by the authors?

13)  Refer to line # 183: Authors need to confirm the title “Smart Industrial Remoting: remoting working in non-digitalised industries”.

14)  Refer to Table 1: Authors have mentioned about weather pattern prediction in environmental observations. Where are the further details?

Good luck.    

Author Response

The Effects of Digitalization for the Sustainability of
Small Farms

Response to Reviewer 1 Comments

1. Summary

Thank you very much for taking the time to review this manuscript. Your critique and suggestions have improved our article considerably. We have added specific improvements to the comments.

2. Questions for General Evaluation

Reviewer’s Evaluation

Response and Revisions

1.Is the content succinctly described and contextualized with respect to previous and present theoretical background and empirical research (if applicable) on the topic?

Must be improved

The manuscript has been revised and updated, with corrections noted in the text.

2. Are all the cited references relevant to the research?

Must be improved

3. Is the research design appropriate?

Must be improved

4. Are the methods adequately described?

Must be improved

5. Are the results clearly presented?

Must be improved

6. Is the article adequately referenced?

7. Are the conclusions thoroughly supported by the results presented in the article or referenced in secondary literature?

Must be improved

Must be improved

3. Point-by-point response to Comments and Suggestions for Authors

Comments 1: The abstract is poorly written. Needs to be rewritten.

Response 1: Thank you for pointing this out. We agree with this comment.

Comments 2: Last paragraph of section 1 should describe structure of the article i.e. missing in the article.

Response 2:  Thank you for pointing this out. In the last paragraph of section 1, we set out the structure of the article.  

We have clarified the definition of a small farm. (Line 220-229)

Comments 3: Refer to line # 48: Poppe and co-authors or Poppe et. al.? Recheck the whole article for such typos.  

Response 3: Thank you for pointing this out. We recheck the whole article.

Comments 4: Refer to line # 57: … attention of policymakers and society towards small-family farming has risen significantly… Include reference here.  

Response 4:  Thank you for pointing this out. We included reference. (Line 68)

Comments 5: Refer to line # 67: Recheck it [17,18,19;20] [41;42,43] for typos.

Response 5:  Thank you for pointing this out. The editor has reviewed the manuscript.

 Comments 6.: Refer to section 1 and 2: There is insufficient discussion on digitization. Authors need to discuss it in more detail as it is the core topic of this study.

Response 6:  Thank you for pointing this out. We have taken your comment into account and made improvements. (Line 34-35)

Comments 7: Refer to whole article: How sustainable development is linked with digitization?

Response 7: This study aims to investigate the links between digitalization and sustainable development in small farm and to identify which economic, social, and environmental criteria best reflect the changes driven by innovation Many authors have explored the link between promotion and sustainability [4,7,9,22,31,48]. We have taken your comment into account and made same improvements. (Line 120-125).

Comments 8: Refer to section 1: What is the criteria to define big farm and small farm?

Response 8:  We have clarified the definition of a small farm. (Line 220-229)

Comments 9:   Refer to line # 127: Authors need to explicitly mention “benefits of suing digital solutions”

Response 9: We have indicated the benefits in lines 157-161, based on the source cited.

Comments 10:  Refer to line # 144: A single sentence?

Response 10:  The text is revised.

Comments 11: Is Patton the name of author or the name of strategy?

Response 11:   Patton the name of the author.

Comments 12:  Refer to line # 168: Include reference for direct-to-consumer business model. Moreover, what are other business model and why this model is selected by the authors?

Response 12:  The text is revised. The model of farm activity is explained in another way. For the case study, we selected a farm that has implemented a pilot digitalization project, and we also analyzed and evaluated which strategies the farm follows.  The case study strategies are clarified. (Line 202-251)

Comments 13:  Refer to line # 183: Authors need to confirm the title “Smart Industrial Remoting: remoting working in non-digitalised industries”.

Response 13:  The text is revised.

Comments 14:  Refer to Table 1: Authors have mentioned about weather pattern prediction in environmental observations. Where are the further details

Response 14:  The semi-structured interview did not cover this issue.

4. Response to Comments on the Quality of English Language

Point 1: Moderate editing of English language required

Response 1:   The editor has reviewed the English language. Certificate of

Editorial Consultancy.

5. Additional clarifications

Reviewer 2 Report

Comments and Suggestions for Authors

This paper is a case study of the effects of digitization on small farm sustainability including the three pillars of sustainability: social, environmental, and economic. This is a relevant question as digital technologies become increasingly integrated into small farming practices and pose uncertainties about sustainability.

There is a blend of passive and active language throughout this paper. I would change passive to active where possible. For example, "The sustainable development paradigm was chosen" vs "We chose the sustainable development paradigm." But this is a minor point and up to the authors' discretion.

Language switches between "digitization" and "digitalization". This should be consistent unless these are different terms, in which case that should be clarified and emphasized.

On Line 185 why is Toolbox capitalized? This is the also the only time toolbox shows up in the body of the text - is it a reference to a broader concept?

In Table 2 why is the environmental column center aligned instead of top-aligned? It makes the numbered rows harder to follow.

The methods section is written in a way that is hard to follow. It seems to be bouncing back and forth between ideas, methods, and processes. The questionnaire and semi-structured interviews appear to be different methods in some places but the same method in other places. For example it talks about the questionnaire on line 200 as its own method, but then on line 208 it talks about collecting data using semi structured interviews and then immediately begins talking about the questionnaire. Was the questionnaire answered in the interviews or was it a separate method? Was the questionnaire a survey or an interview guide for interviews?

In the matrix table of sustainability effects, were participants asked to check boxes on the effects of digitization or were they asked an open ended question that the researchers then coded into those categories?

What does it mean on line 233 that farmers selected 4 economic, 1 social, and 1 environmental outcome? Again, this is unclear because the methods above are unclear on whether this was a fixed-answer questionnaire or open-ended interview. Were they asked to select their six most important outcomes or did they not see the effect of digitization on those other social and environmental outcomes?

It mentions farmers, but how many were interviewed for this project? Did they fill out the matrix in the same way or is the matrix an aggregate? Were there areas of disagreement or difference between interviewed farmers? In the informed consent section it mentions a singular "participant". Was this just one farmer? That should be made clear in the study.

As a case study of a singular farm, the title of this study is misleading as it is not about farms, plural, and generalizing the results of this study beyond a single case would be difficult. This is especially true given the highly subjective nature of the questions asked, and even more so if they were asked to just a single farmer on the one farm. In the conclusion you mention that you studied small farms again - is this one farm or multiple?

Greater transparency including an interview guide included as an appendix would help ensure validity and reliability for this study.

If possible, table 3 should be formatted so "Environmental" is not cut in half.

Why is there limited economic data on small farms that have adopted digital technologies and why were you limited in asking about quantitative questions? It is fine to use interview methods, but these should be guided by your research question, not limitations like these that do not seem thought out or carefully considered. This study has severe limitations, but they are not these - or at least not only these. The biggest limitation as I see it is generalizability.

Overall this seems like an interesting and relevant topic, but without clear methods the findings lose much of their relevance. I would significantly revise and even consider rewriting entirely the methods section for clarity and transparency. I'm also concerned by the lack of care that seems to have gone into the discussion and conclusion section. The results of a singular case study should be interpreted with caution, especially with these subjective measures of sustainability. There is plenty of research to suggest that digitization can have detrimental sustainability impacts, especially for the environment. Concluding that policymakers should use this to inform policy on the use of digital technologies on farms is vague and potentially misleading. What policies would you suggest based on this study?

Comments on the Quality of English Language

There are a few spots where the English is confusing or awkwardly, but no major issues.

Author Response

The Effects of Digitalization for the Sustainability of
Small Farms

Response to Reviewer 2 Comments

1. Summary

Thank you very much for taking the time to review this manuscript. Your critique and suggestions have improved our article considerably. We have added specific improvements to the comments.

2. Questions for General Evaluation

Reviewer’s Evaluation

Response and Revisions

1.Is the content succinctly described and contextualized with respect to previous and present theoretical background and empirical research (if applicable) on the topic?

Must be improved

The manuscript has been revised and updated, with corrections noted in the text.

2. Are all the cited references relevant to the research?

Yes

3. Is the research design appropriate?

Must be improved

Taken into account, explanations are given for specific questions.

4. Are the methods adequately described?

Cant be improved

Taken into account, explanations are given for specific questions.

5. Are the results clearly presented?

cant be improved

Taken into account, explanations are given for specific questions.

6. Is the article adequately referenced?

7. Are the conclusions thoroughly supported by the results presented in the article or referenced in secondary literature?

Can be improved

Must be improved

Taken into account, explanations are given for specific questions.

3. Point-by-point response to Comments and Suggestions for Authors

Comments 1: This paper is a case study of the effects of digitization on small farm sustainability including the three pillars of sustainability: social, environmental, and economic. This is a relevant question as digital technologies become increasingly integrated into small farming practices and pose uncertainties about sustainability.

Response 1:

Comments 2: There is a blend of passive and active language throughout this paper. I would change passive to active where possible. For example, "The sustainable development paradigm was chosen" vs "We chose the sustainable development paradigm." But this is a minor point and up to the authors' discretion.

Response 2:  The editor has reviewed the English language.

Comments 3: Language switches between "digitization" and "digitalization". This should be consistent unless these are different terms, in which case that should be clarified and emphasized.

Response 3:  Thank you for your comment. The text is revised, using the digitalization definition.

Comments 4: On Line 185 why is Toolbox capitalized? This is the also the only time toolbox shows up in the body of the text - is it a reference to a broader concept?

Response 4:  We have corrected the text to take account of the comment. We studied a farm that had participated in a project to develop a toolbox for the digitalization of small businesses. Toolbox is an approach to rethinking the way, how to use technology, to a set of advice or instruments for solving a particular problem.

Comments 5: In Table 2 why is the environmental column center aligned instead of top-aligned? It makes the numbered rows harder to follow.

Response 5:  Layout of the editorial office

 Comments 6. The methods section is written in a way that is hard to follow. It seems to be bouncing back and forth between ideas, methods, and processes. The questionnaire and semi-structured interviews appear to be different methods in some places but the same method in other places. For example, it talks about the questionnaire on line 200 as its own method, but then on line 208 it talks about collecting data using semi structured interviews and then immediately begins talking about the questionnaire. Was the questionnaire answered in the interviews or was it a separate method? Was the questionnaire a survey or an interview guide for interviews?

Response 6:  Thank you for pointing this out. We regret the inaccuracies. We have prepared the interview guide according to pillars, indicators, and processes. During the interview, we asked about the process, how it has changed after digitalization, and how these changes have influenced the results We have taken your comment into account and made improvements. (Line 195-250)

Comments 7: In the matrix table of sustainability effects, were participants asked to check boxes on the effects of digitization or were they asked an open-ended question that the researchers then coded into those categories?

Response 7:  Based on the theory, we have developed a list of sustainability indicators suitable for small farms (Table 2). This list was followed during the interviews. The farmer mentioned some of the indicators as irrelevant and they were not listed in Tables 3 and 4. We asked an open-ended question about each process then coded into those categories. To clarify the interview answers, we have added explanations to the Results section. (Line 316-322)

Comments 8: What does it mean on line 233 that farmers selected 4 economic, 1 social, and 1 environmental outcome? Again, this is unclear because the methods above are unclear on whether this was a fixed-answer questionnaire or open-ended interview. Were they asked to select their six most important outcomes or did they not see the effect of digitization on those other social and environmental outcomes?

Response 8:  We have taken your comment into account and made improvements. (Section of Result)

Comments 9:   It mentions farmers, but how many were interviewed for this project? Did they fill out the matrix in the same way or is the matrix an aggregate? Were there areas of disagreement or difference between interviewed farmers? In the informed consent section it mentions a singular "participant". Was this just one farmer? That should be made clear in the study.

Response 9: The farm studied was the farm with 2 family members, but one of the farmers was interviewed.  We have revised the text.

Comments 10:  As a case study of a singular farm, the title of this study is misleading as it is not about farms, plural, and generalizing the results of this study beyond a single case would be difficult. This is especially true given the highly subjective nature of the questions asked, and even more so if they were asked to just a single farmer on the one farm. In the conclusion you mention that you studied small farms again - is this one farm or multiple?

Response 10:  In carrying out this study, we have taken into account the growing need for applied research, which is important not only for its theoretical insights but also for its socio-economic impact on agriculture. We only partly agree with your point of view that a “case study based on subjective nature of the questions” has no scientific value. The knowledge economy is based on the codification of tacit knowledge, which is built on the experience of the actors. Our study was carried out with the unique opportunity to interview the owner of a farm that has adopted digital technologies. It was also important that we had reliable information about the digital technologies and the small farm's processes. We accept that the experience of one farm as a subject of the study may raise questions about the applicability of the findings to other similar farms. For this reason, we have focused on the theoretical validity of the use of the case study approach, and we have drawn on other authors' statements on the use of case studies to investigate new and original phenomena.  We agree with your observation that the findings are not appropriate for all farms and have therefore clarified the group of farms for which the findings are appropriate according to the case study selection strategies. (Line 250-252; 419-421).

Comments 11: Greater transparency including an interview guide included as an appendix would help ensure validity and reliability for this study.

The structure of the interview guide is designed as a matrix of processes and indicators, in which one component is made up of the processes shown in lines 209-216, and the other component is made up of the indicators in Table 2.

Comments 12:  Why is there limited economic data on small farms that have adopted digital technologies and why were you limited in asking about quantitative questions? It is fine to use interview methods, but these should be guided by your research question, not limitations like these that do not seem thought out or carefully considered. This study has severe limitations, but they are not these - or at least not only these. The biggest limitation as I see it is generalizability.

Response 12:  In the case of our study, we asked farmers to measure the effect of how processes are organized on the farm, rather than information at the farm level.  Process-based accounts are not available, so farmers cannot provide such accounting data.  On the other hand, there are no publicly available farm economic statistics that would allow a comparison of the results of digitized and non-digitized farms, or the dynamics of the results of digitized farms. The main public source that provides information on the economic performance of farms in EU countries is FADN, which does not collect data on the digitalization characteristic. That is the reason, why we have not been able to find a study that analyses the economic indicators of farms using digital technologies.

Comments 13:  Overall this seems like an interesting and relevant topic, but without clear methods the findings lose much of their relevance. I would significantly revise and even consider rewriting entirely the methods section for clarity and transparency. I'm also concerned by the lack of care that seems to have gone into the discussion and conclusion section. The results of a singular case study should be interpreted with caution, especially with these subjective measures of sustainability. There is plenty of research to suggest that digitization can have detrimental sustainability impacts, especially for the environment. Concluding that policymakers should use this to inform policy on the use of digital technologies on farms is vague and potentially misleading. What policies would you suggest based on this study?

Response 13:  Thank you sincerely for raising the discussion points that have helped us to substantially improve our article. Changes have been made to all sections of the article.

4. Response to Comments on the Quality of English Language

Point 1: There are a few spots where the English is confusing or awkwardly, but no major issues.

Response 1:  The editor has reviewed the English language: Certificate of Editorial Consultancy.

5. Additional clarifications

Reviewer 3 Report

Comments and Suggestions for Authors

In the study, the authors addressed the interesting problem of the impact of digitalization on the issues of sustainable development in agricultural farms. The strong point is the originality of the problem, but the part related to the discussion and conclusions should be improved.

Major comments

Clearly indicate the purpose of the research. However, the search for benefits itself is a bit of an unambitious task. The problem with digitalization concerns the barriers and profitability of use in small business entities.

Can the farm selected as a case study really be considered a typical farm as the authors indicate? This requires appropriate justification. The size of the farm (0.5 ha) does not favor the employment of external labor.

What forms (systems) of digitization were used in the surveyed entities? The results indicate very broad effects of activities for small entities.

It is better to clearly distinguish the conclusions as a separate section.

The conclusions are too general and often refer to the organization of the research itself by the authors. Indicate clearly what results from the research conducted. Recommendations for decision-makers would also be of significant value.

Minor comments

The summary should indicate what the purpose of the study was and what the most important results were obtained.

Unification of bibliographic records. Once in the page range there is one p. (e.g. item 16) and once in p. (item 10). It happens that the entry appears without the letter p. (item 26).

Author Response

The Effects of Digitalization for the Sustainability of
Small Farms

Response to Reviewer 3 Comments

1. Summary

Thank you very much for taking the time to review this manuscript. Your critique and suggestions have improved our article considerably. We have added specific improvements to the comments.

2. Questions for General Evaluation

Reviewer’s Evaluation

Response and Revisions

1.Is the content succinctly described and contextualized with respect to previous and present theoretical background and empirical research (if applicable) on the topic?

Can be improved

Taken into account, explanations are given for specific questions.

2. Are all the cited references relevant to the research?

Yes

3. Is the research design appropriate?

Must be improved

Taken into account, explanations are given for specific questions.

4. Are the methods adequately described?

Must be improved

Taken into account, explanations are given for specific questions.

5. Are the results clearly presented?

Must be improved

Taken into account, explanations are given for specific questions.

6. Is the article adequately referenced?

7. Are the conclusions thoroughly supported by the results presented in the article or referenced in secondary literature?

Can be improved

Must be improved

Taken into account, explanations are given for specific questions.

3. Point-by-point response to Comments and Suggestions for Authors

Comments 1: Clearly indicate the purpose of the research. However, the search for benefits itself is a bit of an unambitious task. The problem with digitalization concerns the barriers and profitability of use in small business entities.

Response 1: Thank you for pointing this out. We agree with this comment. (Line 93-98.)

Comments 2: Can the farm selected as a case study really be considered a typical farm as the authors indicate? This requires appropriate justification. The size of the farm (0.5 ha) does not favor the employment of external labor.

Response 2:  Thank you for pointing this out.  We have clarified the definition of a small farm. (Line 220-229)

Comments 3: What forms (systems) of digitization were used in the surveyed entities? The results indicate very broad effects of activities for small entities.

Response 3:  The farm has implemented a digitalization pilot project and a digital management system with several modules, which is the reason why there is such a wide-ranging impact on activities.

Comments 4: It is better to clearly distinguish the conclusions as a separate section. Recommendations for decision-makers would also be of significant value.

Response 4:  Thank you for pointing this out. In preparing the article, we have followed the editorial-recommended paragraph structure. We have presented the conclusions together with the discussion, as this design helps to better highlight the findings and limitations of the study.

Comments 5: The summary should indicate what the purpose of the study was and what the most important results were obtained.

Response 5:  Thank you for pointing this out. We have taken note of your comment and revised the summary.

 Comments 6. Unification of bibliographic records. Once in the page range there is one p. (e.g. item 16) and once in p. (item 10). It happens that the entry appears without the letter p. (item 26).

Response 6:  Thank you for pointing this out. We have taken your comment into account and made improvements.

Comments 7: The conclusions are too general and often refer to the organization of the research itself by the authors. Indicate clearly what results from the research conducted. Recommendations for decision-makers would also be of significant value.

Response 6:  Thank you for pointing this out. We agree with the comment that the conclusions are general, as the specific results of the study are presented in the Results section. Also, results from the research are conducted on lines 380-406. Depending on the comment, we have added recommendations for policymakers. (Line 417-422)

4. Response to Comments on the Quality of English Language

Point 1: English language fine. No issues detected

Response 1:  

5. Additional clarifications

Round 2

Reviewer 1 Report

Comments and Suggestions for Authors

Dear Author(s),

My comments are addressed and I have no more comments.

Good luck.    

Comments on the Quality of English Language

Dear Author(s),

My comments are addressed and I have no more comments.

Good luck.    

Reviewer 2 Report

Comments and Suggestions for Authors

Thank you for taking the time to address my previous comments.

Although the additions have added some clarity on the points I raise, I do think that the same issue continue to be present. This paper has done a lot of work to polish very weak data, and the results continue to overstate the generalizability and findings from an interview with a single farmer, who may not even be very representative of the farming population - although this is hard to tell because the study site is not clearly defined. I can't see in which country this study took place. There are some concepts that are okay to generalize and say within the EU. A single farm in Lithuania (if that is where this study took place) may run into very different issues than a farm in other EU issues, and this is another area where this study is not clear with the methodology.

I think this is a promising pilot study, but it needs more research and data.

Comments on the Quality of English Language

It seems to be more grammatically correct that you should refer to the farmer as, well, "the farmer" rather than "farmer". E.g. "Farmer said that the introduction of digitization..."

You also refer to the farmer as a woman but use the pronoun "his", which should be "her".